# Exploring the p53 connection of cervical cancer pathogenesis involving north-east Indian patients

**Mohammad Aasif Khan**[1], **Diptika Tiwari**[2], **Anita Dongre**[1], **Sadaf**[1], **Saad Mustafa**[1], **Chandana Ray Das**[3], **Sheersh Massey**[1], **Purabi Deka Bose**[4], **Sujoy Bose**[2], **Syed Akhtar Husain**[1] *

**1** Human Genetics Laboratory, Department of Biosciences, Jamia Millia Islamia, New Delhi, India, **2** Department of Biotechnology, Gauhati University, Guwahati, Assam, India, **3** Department of Obstetrics & Gynecology, Gauhati Medical College and Hospital, Guwahati, Assam, India, **4** Department of Molecular Biology and Biotechnology, Cotton University, Guwahati, Assam, India

* shusain@jmi.ac.in, akhtarhusain2000@yahoo.com

**Data Availability Statement:** All relevant data are within the paper and its Supporting Information files.

## Abstract

### Background

As per WHO, Cervical cancer (CaCx) is a global issue, being the fourth common cancer in women with incidence rate of 13.1 per 1 lakh women globally and accounting for 311000 deaths in the year 2018 itself globally. The molecular pathogenesis in Human papillomavirus (HPV) infected cases is inconclusive. The detection of molecular factors leading to progression of CaCx can be important in the diagnosis and management of the disease. p53 a known tumor suppressor gene having a regulative role in cell cycle has been highlighted as key factor in the prevention of cancer but its significance in CaCx cases has been variably documented. The present study therefore targeted to evaluate the significance of p53 profile in CaCx cases in ethnically distinct northeast Indian population.

### Methods

Blood and Tissue samples (N = 85) of cervical cancer patients were collected and screening for HPV was performed using PCR. Thereafter the differential mRNA expression(qPCR), Immunohistochemistry, Mutation (PCR direct sequencing method) of p53 was studied. Further p53 epigenetic profiling was done by Methylation specific PCR (MS-PCR) and western blotting by using p53 acetylation specific antibodies.

### Results

Our findings revealed that the downregulation of p53 was associated with the progression of disease and the variation in downregulation based on p53 polymorphism was observed. Further hypermethylation and deacetylation of p53 was also found to be associated with the pathogenesis of CaCx. The downregulated expression and hypermethylation of p53 in lower grade of CaCx, together established its association with the progression of CaCx from lower to severe grade.

**Funding:** yes This work was supported by the Department of Biotechnology, Ministry of Science and Technology (BT/501/NE/TBP/2013).

**Competing interests:** The authors have declared that no competing interests exist.

**Abbreviations:** CaCx, Cervical Cancer; WHO, World health Organisation; HPV, Human Papilloma Virus; MS-PCR, Methylation specific PCR; HDAC, Histone deacetylase; qPCR, Quantitative PCR.

## Conclusion

Therefore, in CaCx patients of northeast Indian population, malfunctioning of p53 is found to have significant role in cervical cancer progression.

## Introduction

In women, cervical cancer (CaCx) ranks as the fourth most common cancer, and among all the cancer it ranks as the seventh most common cancer occurring worldwide [1]. It is **a** leading cause of cancer related mortality in Indian women of both rural and urban areas [2]. The occurrence of the cervical cancer in woman is increasing annually [3]. The development of cervical cancer has been associated with the HPV infection which is one of the established factors associated with the occurrence of cervical cancer lesion [4]. The difference in the clinical presentation, the progression of the disease as well as response to treatment differs amongst individuals both with and without underlying HPV infection, thereby suggesting the role of specific host genetic factors in the pathogenesis of the disease. Identifying these key factors at molecular level may prove to be a vital link in the approaches required for controlling the disease and/or establishing the significance of molecular markers indicating the prognosis of the disease.

The TP53, a tumor suppressor gene which is also known as the guardian of the genome or the cellular gatekeeper of growth and division, located on chromosome 17pl3.1 is involved in the cell proliferation, regulation of cell cycle, DNA repair, promoting apoptosis, suppressing angiogenesis and migration of tumor cells thus preventing metastasis [5–7]. The inactivation and malfunctioning of p53 gene are found to be associated with the development and progression of many human cancers including cervical cancer [6]. HPV viral onco-proteins are known to regulate host tumour suppressor proteins such as p53 which result in the malfunctioning of tumor suppressor protein [8]. The data in the role of deregulated p53 in CaCx pathogenesis has been suggested, but differences exists in the data documented from different geographical niches; which underlines the importance of understanding the molecular diversity in the importance of p53 in specific population context [9].

The diversity in the p53 link of any disease pathogenesis including cancers of different aetiologies is dependent on the genetic and epigenetic regulation of p53 gene, which includes transcriptional regulation of p53 gene expression, genetic alterations as well as epigenetic regulations such as its methylation and acetylation profile [6, 10]. In human cancer TP53 is regarded as one of the most common mutated genes which generally involve the single base substitution mutation in the DNA binding domain that changes the function of the protein [11].

Epigenetic changes have been demonstrated to play an important role in cancer pathogenesis. The common epigenetic changes in human being are found to be DNA methylation and posttranslational histone protein modifications such as methylation, acetylation etc. [12]. Inactivation of p53 has been reported to be due to hyper-methylation of the promoter region [13] and it has been associated with human neoplasia [14]. Acetylation occurring in the lysine residue of histone peptides has been found to impart changes in the expression pattern of the genes [15]. Acetylation is known to have critical effects on p53, as it increases p53 protein stability, binding to low affinity promoters, association with other proteins, antiviral activities, and is required for its checkpoint responses to DNA damage and activated oncogenes [10, 16, 17].

Although the importance of genetic alterations in p53 gene has been reported to be associated with cancer of other aetiologies like stomach cancer [18], lung cancer [19] and nasopharyngeal carcinoma [20] in northeast Indian population, but no reports exists for its association with CaCx pathogenesis in the ethnically distinct northeast Indian population majorly comprising population of tribal dominance eg Bodos, Kacharis, Karbis, Dimasas, Tiwa and Lalungs etc. Presented herein is a study involving northeast Indian population for the significance of p53 signatures in the pathogenesis of CaCx.

## Materials and methods

### Enrolment of patient and sample collection

For the present study the clinically diagnosed cervical cancer patients were enrolled. Both blood (3ml) and tissues were collected from patients (n = 85) with written informed consent by the Gynaecologist of Obstetrics and Gynaecology department, Gauhati Medical College and Hospital. The tissue biopsies were obtained from patients for both cancerous and adjacent non-neoplastic region. Cases documented with any pathogenic infection apart from HPV, and any subjects not providing written informed consent were excluded from the present study. This study was approved by the ethical committee of Jamia Millia Islamia, Cotton University, Gauhati Medical College and Hospital Gauhati University. (Reference No. GUIEC-3990/2014, GMCH: MC/06/2014/53)

### Processing of samples

The collected tissue samples were snap freezed in liquid nitrogen (for DNA and protein-based work), in RNA later vials stored at -80˚C (for RNA based study) and a portion of tissue sample was fixed in 10% formalin and embedded in paraffin for histopathological examination and immunohistochemistry-based analysis at protein level.

### DNA extraction

Total DNA was isolated from both affected and the adjacent non-affected region of cervix, by treating the samples with tissue lysis buffer and proteinase K digestion, followed by standard phenol-chloroform based extraction. This cellular DNA was further precipitated with the help of an alcohol and finally suspended in adequate amount of TE buffer.

### HPV screening and genotyping by PCR

The established MY09/11 primers (F:5′CGTCCMARRGGAWACTGATC3′ R:5′GCMCAGGGW CATAAYAATGG3′) were used to confirm the presence of HPV DNA. Further **HPV** 16 and HPV 18 screening for the collected sample was done by PCR analysis using the specific primers for HPV16 and HPV18. The primers sequences used for HPV 16 were: F:5′TCAAAAGCCA CTGTGTCCTGA3′ and R: 5′GGTGTTCTTGATGATCTGCAA3′; whereas the primer sequences used for the HPV 18 screening were: F: 5′CCGAGCACGAC AGGAACGACT3′ and R:5′TCGTT TTCTTCCTCTGAGTCGCTT3′.

Further the blood samples were used to screen the presence of any other pathogenic infection such as HIV, Hepatitis's virus apart from HPV in the collected samples. The screening for HIV, HBV, HCV, HAV and HEV was performed using ELISA kit (Erba Lisa HIV Gen4, ErbaLisa SEN HBsAg, ErbaLisa HCV Gen 3 (v2), MP DIAGNOSTICS (MPD) HAV IgM ELISA 4.0 and MP Diagnostics HEV ELISA 4.0; US)

## TP53 expression analysis

The differential expression of p53 was studied both at mRNA and protein level by Real-time PCR and immunohistochemistry respectively. For differential mRNA expression analysis, total RNA was isolated by trizol (Qiagen, US) method using manufacturer protocol, converted to cDNA using commercially available cDNA synthesis kit (Applied Biosystems, US) thereby following the manufacture protocol. The cDNA template was used for the differential expression analysis by Real-time PCR (Applied Biosystem 7500 Fast) using SYBR-Green chemistry and primers specific for p53 (F: 5′ TACTCCCCTGCCCTCAACAA 3′ and R: 5′ CATCGC‐TATCTGAGCAGCGC 3′]. The *β-actin* (F: 5′AGATAGTGGATCAGCAAGCAG3′ and R: 5′ GCGAAGTTAGGTTTTGTCA3′) expression was used as internal normalization control for fold change analysis using the $2^{-\Delta\Delta Ct}$ method. The real time PCR master mix was prepared for 20 μl of reaction and PCR cycling condition were set as: initial denaturation at 95˚C for 5 minutes followed by 40 cycles of 94˚C for 30 secs, 58˚C for 30 secs and 72˚C for 30 secs. The data was acquired at 72˚C for 45 secs.

The p53 differential protein expression was studied by immunohistochemistry analysis using antibody specific for p53 (ab26, *Abcam*, UK) after deparaffinization, antigen retrieval by boiling in citrate buffer for rounds of 2mins at full power in a microwave oven followed by alcohol dehydration in different grades of alcohol percentage, xylene wash and overnight incubation with antibody specific for p53 at 1:1000 dilution. The detection was done using the super sensitive one-step polymer-HRP detection system (*Biogenex, US*). The slides were finally examined and graded for P53 expression by a senior pathologist. The expression was graded as strong (+++), moderate (++), low (+) or no expression (-) in all the studied cases and controls.

## *TP53* mutation analysis

In human cancer p53 is regarded as one of the most commonly altered genes. Although more than 200 single nucleotide polymorphisms have been identified (http://www-p53.iarc.fr/), SNPs within exons 4–9 are of critical relevance because of their positioning in the DNA binding domain, the presence of which can change the function of the protein [11]. Thus, genetic alterations of the p53 gene (exon 4–9) was screened in all the subjects by PCR-direct sequencing method using specific primers (Table 1) for its association with CaCx pathogenesis in ethnically distinct Northeast Indian population. The PCR amplification was done for 30 μl of reaction using readymade PCR master mix 2X (Genei^TM, India) following the manufacturer protocol. PCR cycling condition were set as: initial denaturation at 95˚C for 5 minutes followed by 40 cycles of 94˚C for 30 secs, Annealing (temperature provided in table for each exon) for 30 secs, extension at 72˚C for 30 secs followed by a final extension at 72˚C for 7 minutes. The amplified product after confirmation via agarose gel electrophoresis was sent for

**Table 1. List of primer sequence of *p53* gene from exon 4–9 and their annealing temperature.**

| S.No | Exon | Forward primer 5′-3′ | Reverse primer 5′-3′ | Tm˚C |
|---|---|---|---|---|
| 1. | 4 | AATGGATGATTTGATGCTGTCCC | GCCAAGTCTGTGACTTGCACG | 59 |
| 2. | 5 | GCCAACTCTCTCTAGCTCGC | GATAGCGATGGTGAGCAGCT | 58 |
| 3. | 6 | CCTCATCTTGGGCCTGTGTT | CACCTCTCATCACATCCCCG | 61 |
| 4. | 7 | TGGGAGTAGATGGAGCCTGG | AGGGAGCACTAAGCGAGGTA | 58 |
| 5. | 8 | CCTCTTTCCTAGCACTGCCC | GGGCAGTGATGCCTCAAAGA | 61 |
| 6. | 9 | CAATGGCTCCTGGTTGTAGC | CACCTAATCTAAGGAACATCATA | 57 |

direct sequencing. The difference in distribution of specific SNP was evaluated statistically using SPSSv13.0 software.

## TP53 epigenetic profiling

As the majority of the cases were of underlying HPV etiology, for the epigenetic study we aimed to focus on the paired HPV positive cases (i.e. cases with both cancerous and non-cancerous region) specifically for data specificity.

[A] P53 methylation assay. Difference in p53 Promoter methylation profiling was studied by methylation specific PCR method (MSP method). Briefly, the DNA extracted from the HPV positive paired samples were subjected to bisulphite conversion using EZ DNA Methylation- Gold Kit (Zymo Research, *US*) following the protocol as instructed in the kit. The bisulphite converted DNA was subjected to MSP amplification using specific sets of TP53 methylated primers (MF:5'-TTGGTAGGTGGATTATTTGTTT-3'; MR 5'-CCAATCCAAA AAAACATATCAC-3') and TP53 unmethylated primers (UF: 5'-TTCGGTAGGCGGATTAT TTG-3', UR: 5'-AAATATCCCCGAAACCCAAC-3') separately and evaluated finally in amplification pairs by agarose gel electrophoresis. The methylated primers amplified a product size of 250bp where as the unmethylated primers amplified a product size of 120bp.

[B] P53 acetylation study. The acetylation profile of p53, which is associated with p53 transcript stability and in turn influencing p53 protein expression, was studied for specific acetylation sites K305 (ab109396), K373 (ab62376) and K382 (ab75754) with p53-acetylation specific antibodies (Abcam, UK) by western blot analysis (Ac-305) and by immunohistochemistry (Ac-373 and Ac-382) following standard protocols.

For western blot analysis the protein was isolated from the both cancerous and noncancerous region of the cervix using Magnesium lysis buffer (MLB) and was quantified by BSA (Bovine serum albumin) method. After quantification 50μg of protein was used to perform SDS-PAGE (Sodium dodecyl sulphate-polyacrylamide gel electrophoresis). After SDS-PAGE the immunoblotting was perform using nitrocellulose membrane (HYBOND) and the transfer efficiency of protein was checked by visualising Ponceau H stain. Further the membrane was blocked using a blocking reagent followed by washing and overnight incubation with the specific antibody (1:1000 for Antip53, Ac-305). After overnight incubation the membrane was washed thoroughly and developed using Chemiluminescence ECL prime Amersham, UK) developing kit from GE Healthcare following manufacturer's protocol. The Chemiluminescence based results were captured by exposing the X-ray film to blot for specific time interval. The film were analysed for target protein differential expression in normal light. Total *β-actin* expression was used as internal control for western blot protein expression analysis.

The immunohistochemistry based assay of p53 acetylation profile Ac-373 and Ac-382) was studied after deparaffinization, antigen retrieval by boiling in citrate buffer for rounds of 2 mins at full power in a microwave oven followed by alcohol dehydration in different grades of alcohol percentage, xylene wash. The slides were incubated overnight with p53 specific acetylation antibodies (1:100 for K373 and 1: 250 for K382). The detection was done using the super sensitive one-step polymer-HRP detection system (*Biogenex, US*). The slides were finally examined and graded for P53 expression by a senior pathologist.

## Statistical analysis

All statistical analyses were performed by the standard methods using SPSS software v13, (SPSS Inc., Chicago, IL, USA). Results were expressed as means± standard deviations (SD). The significance was described as Pearson p-value and tailed p-value less than 0.05 was considered statistically significant.

## Results

### Demographical and clinical profile of the enrolled patients

In the present study, cervical tissues and blood samples were collected from women (N = 85) suffering from cervical cancer. The samples were collected from women between the age group of 18–60 years. In our study we found that the majority of collected cases belonged to the age group of 31–40 years followed by 41–50 and 51–60 age groups being same, indicating the higher prevalence of cervical cancer in the early and reproductive age group of females. In cervical cancer patient the mean gravida in all the age group was found to be 4.59. Based on the screening report, out of 85 collected samples, 84.7% (N = 72) of cervical cancer cases were found to be HPV positive (Table 2). All the HPV cases were HPV16 positive, and none of the cases were positive for HPV18. The ELISA based assay for the presence of HIV, HAV, HBV, HCV, and HEV infection showed negative results for all the enrolled cases.

### P53 expression profile analysis

The *p53* differential mRNA expression in cancerous area compared to paired non-neoplastic adjacent area was evaluated by Real-time PCR using specific primers and SYBR-Green chemistry and *β-actin* as internal normalization control. The p53 mRNA expression in affected area was down regulated in both HPV positive cases (fold change = 0.624 ± 0.312 folds) as well as non-HPV infected cases (fold change = 0.463 ± 0.218 folds), underlying the importance of downregulation of p53 in the development of CaCx (Fig 1).

 Since majority of the cases had an underlying HPV infection (84%), therefore the differences in *p53* mRNA expression in different stages of CaCx was further compared. The

**Table 2. Demographical and clinico-pathological profile of cervical cancer subjects in the study.**

| Characteristics | Number of cases [%age] | |
|---|---|---|
| | **HPV positive (N = 72) [84.70]** | **HPV negative (N = 13) [15.29]** |
| *Age group* | | |
| 18–30 years | 08 [11.11] | 01 [7.69] |
| 31–40 years | 26 [36.11] | 08 [61.53] |
| 41–50 years | 19 [26.38] | 02 [15.38] |
| 51–60 years | 19 [26.38] | 02 [15.38] |
| *Menopausal status* | | |
| Pre-Menopausal | 44 [61.11] | 10 [76.92] |
| Post-Menopausal | 28 [38.88] | 03 [23.07] |
| *Histological type* | | |
| Well differentiated SCC | 43 [59.72] | 07 [53.84] |
| Moderately differentiated SCC | 21 [29.16] | 04 [30.76] |
| Poorly differentiated SCC | 08 [11.11] | 02 15.38] |
| *Stage* | | |
| IIA | 32 [44.44] | 05 [38.46] |
| IIB | 16 [22.22] | 04 [30.76] |
| IIIA | 09 [12.5] | 02 [15.38] |
| IIIB | 11 [15.27] | 02 [15.38] |
| IV | 04 [5.55] | 00 [0.00] |
| *Gravida* | 4.59 (Mean) | |
| **HPV genotype** | | NA |
| HPV16 | 72 [100.00] | |
| HPV18 | 00 [0.00] | |

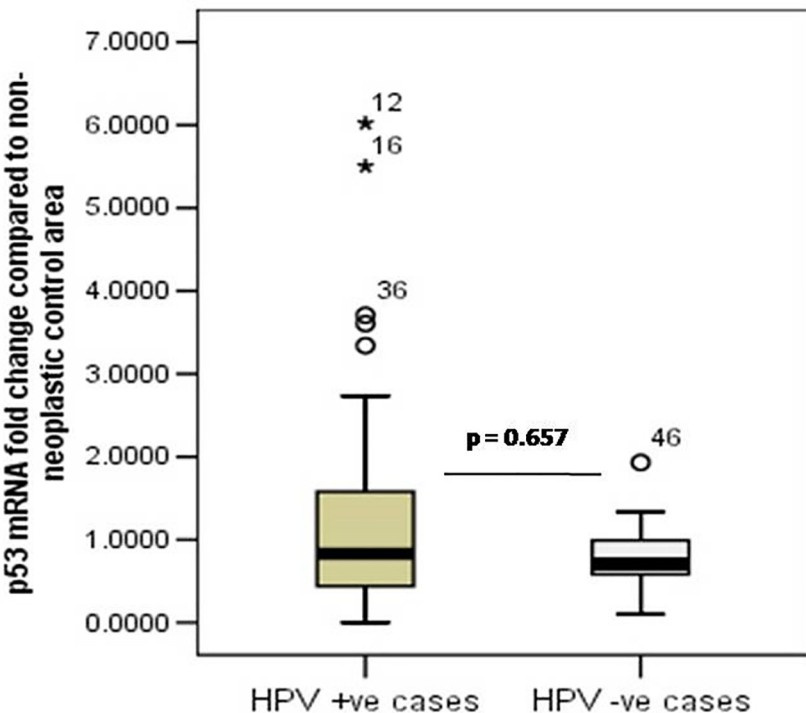

**Fig 1. Box plot analysis showing the downregulation of p53 mRNA expression in HPV positive and HPV negative cervical cancer cases.**

downregulation of *p53* mRNA expression was not uniform in CaCx cases of different severity grades. The *p53* mRNA expression was downregulated in lower severity stage (stage IIA and IIB) compared to higher stages of CaCx (≥stage III) (Fig 2A). The expression of *p53* was

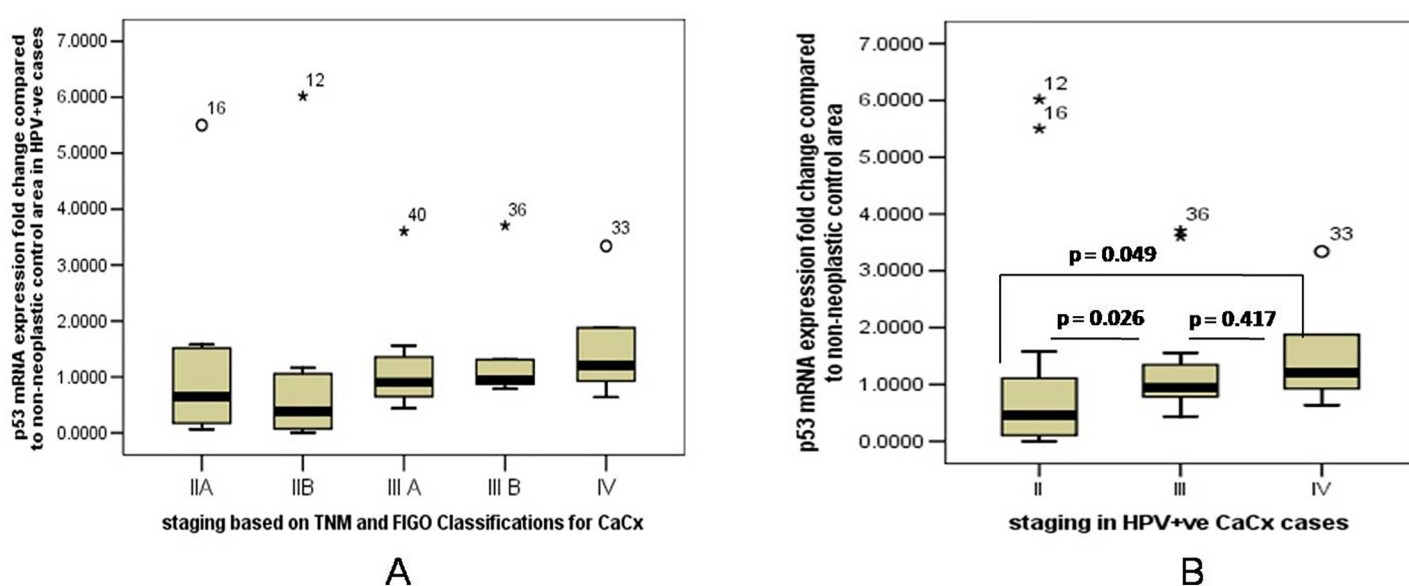

**Fig 2.** [A] Box plot analysis showing differential p53 mRNA expression in HPV positive cases showing downregulation in lower severity stage (stage IIA and Stage IIB) compared to higher stages of Cervical cancer (≥stage III) as per the TNM and FIGO staging of cervical carcinoma. [B] Box plot showing differential p53 mRNA expression analysis in stage II III and IV of HPV positive cervical cancer cases.

significantly downregulated in stage II compared to stage III (p = 0.026) and stage IV (p = 0.049) also p53 expression was downregulated in stage III compared to stage IV (Fig 2B). The p53 mRNA expression was also downregulated in well differentiated CaCx cases compared to both moderately differentiated (p = 0.107) and poorly differentiated cases (p = 0.062).

The differential p53 protein expression was analysed by immunohistochemistry in HPV positive cases. Histological slides of both cancerous and non-neoplastic control region were analysed, and the data was found to be consistent with the p53 mRNA expression data for HPV cases. The protein level expression of p53 was down-regulated in the cancerous region compared to non-neoplastic adjacent control region in majority of the cases studied (43/72, 59.72%). Similar to mRNA expression the expression of p53 protein was down-regulated in lower severity grade ($\leq$ stage III) compare to higher severity grade ($\geq$stage III) indicating its role in the initial stage of development of cervical cancer (Fig 3).

## Mutational analysis of p53 gene in exons (4–9) and their association with cervical cancer pathogenesis

To evaluate the role of altered p53 gene in the susceptibility to CaCx, the mutation study for six different exons (exon4→9) for p53 gene was done. For the mutation study the PCR amplification for six different exons was performed using the specific primers and then the amplified PCR product was sent for sequencing to assess the presence of mutation in all the six exons (Fig 4A). The analysis result from the sequencing showed changes in only exon4 of the p53 gene in all the enrolled cases (rs1042522: Pro72Arg) (Fig 4B, S1 Table). All the three genotypes i.e. Pro/Pro [HPV+ve = 14/72 (19.44%), HPV-ve = 7/13 (53.84%)], Arg/Pro [HPV+ve = 39/72 (54.16%), HPV-ve = 4/13 (30.77%)], and Arg/Arg genotypes [HPV+ve = 19/72 (26.38%), HPV-ve = 2/13 (15.38%)] were observed in the studied cohort.

The changes in mRNA fold change profile based on genotype differences were hence evaluated. The analysed data indicated that there is a difference in the mRNA expression profile pattern in the Pro/Pro vis-à-vis Arg/Pro heterozygous or Arg/Arg homozygous genotype in HPV positive and HPV negative cases. While an upregulation of p53 mRNA fold changes was observed in Arg/Pro

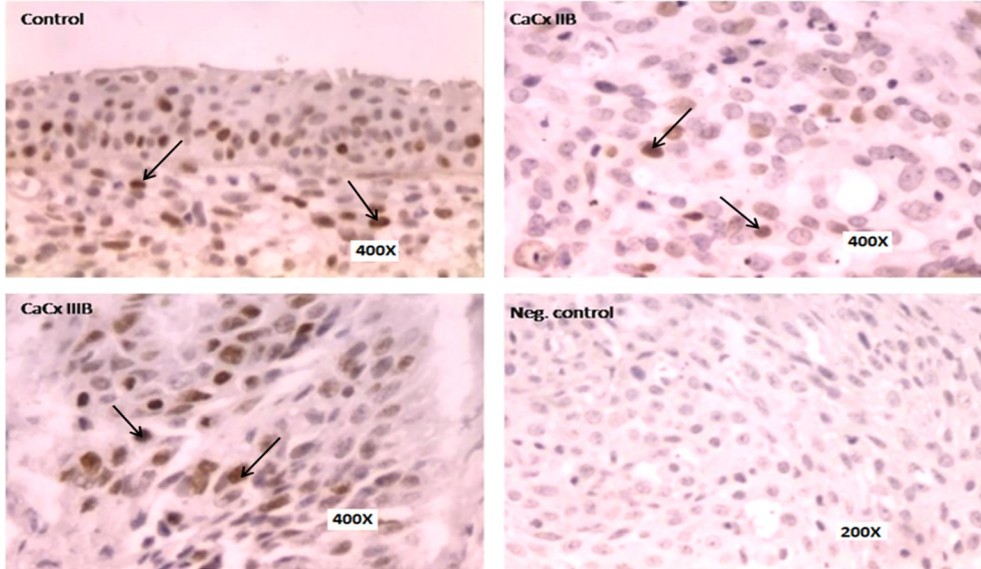

**Fig 3. Representative panel of IHC results showing downregulation of p53 expression in HPV related cervical cancer cases especially in lower grade of CaCx cases (stage II) compared to higher grade (stage III).**

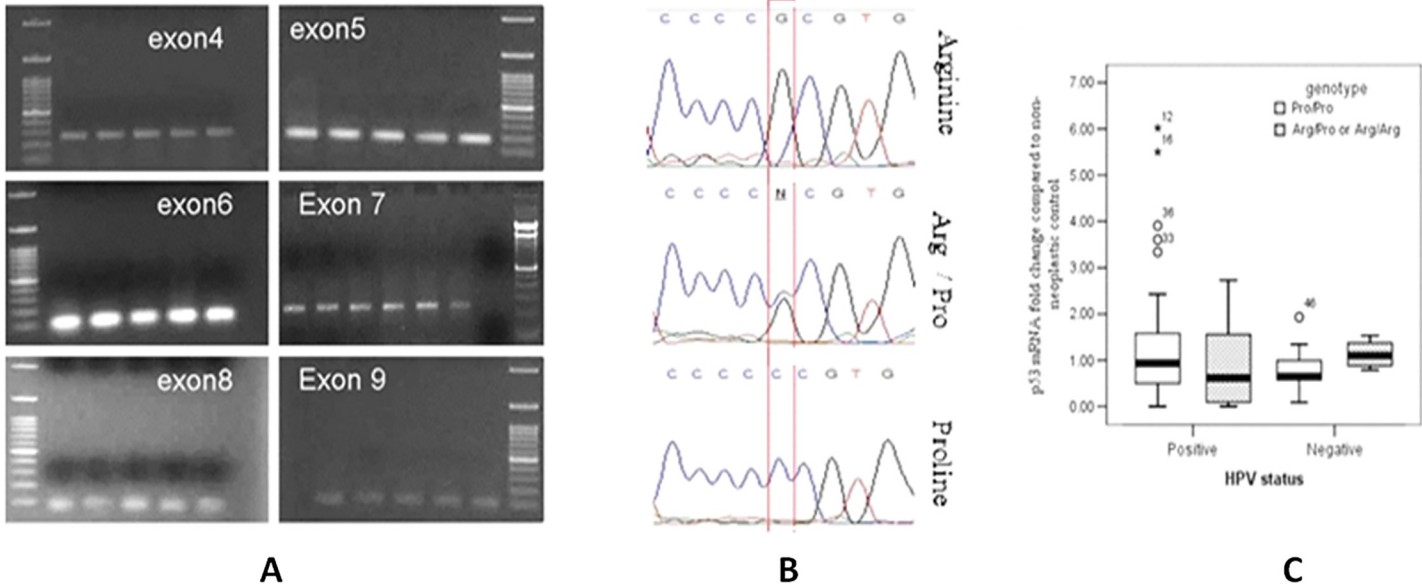

**Fig 4.** [A] Representative agarose gel electrophoresis photograph showing the PCR amplification of exon 4,5,6,7,8 and 9 of p53 gene for mutation analysis in cervical cancer cases. [B] Representative electrophoregrams showing the presence of Arg/Arg, Arg/Pro heterozygous and Pro/Pro genotype based on p53 exon 4 sequencing analysis in the studied Cervical cancer patient population cohort, [C] Box-plot analysis showing the difference in pattern of p53 mRNA expression fold change profile in Arg/Pro heterozygous or Arg/Arg homozygous genotype compared to the Pro/Pro genotype in HPV+ve and HPV-ve cases, thereby signifying the importance of rs1042522 polymorphism in the northeast Indian population.

heterozygous or Arg/Arg homozygous genotype compared to the Pro/Pro genotype CaCx cases in HPV–ve cases (p = 0.362), it was an opposite scenario in HPV +ve cases where a down-regulated p53 expression was observed in the Arg/Pro heterozygous or Arg/Arg homozygous genotype cases compared to the HPV+ve CaCx cases with Pro/Pro genotype (p = 0.297) (Fig 4C).

## Epigenetic profile of p53 gene and its association with cervical cancer pathogenesis

[A] **P53 promoter methylation.** Difference in p53 Promoter methylation profiling was studied by methylation specific PCR method (MSP method) (Fig 5). The data achieved was

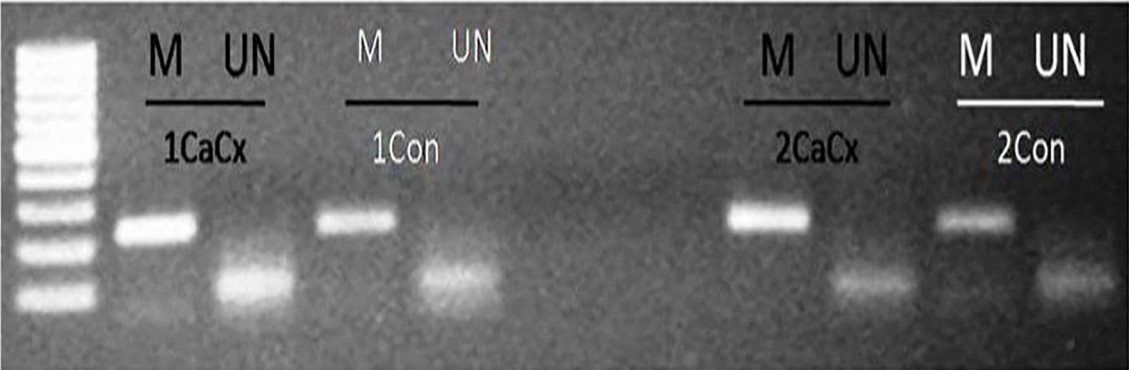

**Fig 5.** Representative agarose gel electrophoresis photograph showing p53 methylation profile analysis by MSP methods in Cacx (cervical cancer) and Con (non neoplastic control) cases; M represents methylated and UN represents unmethylated based assay.

**Table 3. Methylation status in affected area compared to non-neoplastic control area in HPV positive cases.**

| Staging of Cervical Cancer cases | N | Hyper-methylation | Hypo-methylation | No change |
|---|---|---|---|---|
| Grade II | 31 | 12[38.7] | 5 [16.1] | 14[45.1] |
| Grade III | 27 | 7[25.92] | 9[33.33] | 11[40.74] |
| Grade IV | 14 | 1[7.14] | 5[35.71] | 8[57.14] |

heterogeneous with hyper-methylation (N = 20/72, 27.77%), hypo-methylation (N = 19/72, 26.38%) and no alteration (N = 33/72, 45.83%) in methylation levels being observed in the CaCx affected area compared to non-neoplastic adjacent area of the paired HPV infected cases evaluated in the study (Table 3), suggesting that alteration in the promoter methylation profile of p53 may have an influential role to play in the pathogenesis in a sub-cohort of CaCx cases by regulating the p53 mRNA expression (Fig 4). When we considered the cases where changes in p53 promoter methylation was observed, the differences in methylation profile indicated that p53 promoter hyper-methylation was significantly associated lower severity grade (stage IIA and IIB) compared to higher severity grade ($\geq$stage III) of CaCx cases (p = 0.036); which correlates with the expression pattern of P53 at mRNA and protein level, and underlines the significance of methylation in p53 biology in CaCx.

**[B] TP53 acetylation and cervical cancer.** Acetylation also plays a key role in p53 protein stabilization and transcriptional regulations. The expression of Ac-305 and Ac-382 p53 protein was consistently downregulated in majority of the HPV infected CaCx cases. Comparatively, in the same cases the p53 K-373 expression showed stable expression (Fig 6). Acetylation of K305 and K382 holds key for the p53 stabilization and transcription activity, and hence the present data underlines the significance of loss of acetylation at key K305 and K382 sites in p53 and suggests that it may play a role in CaCx pathogenesis in northeast Indian population.

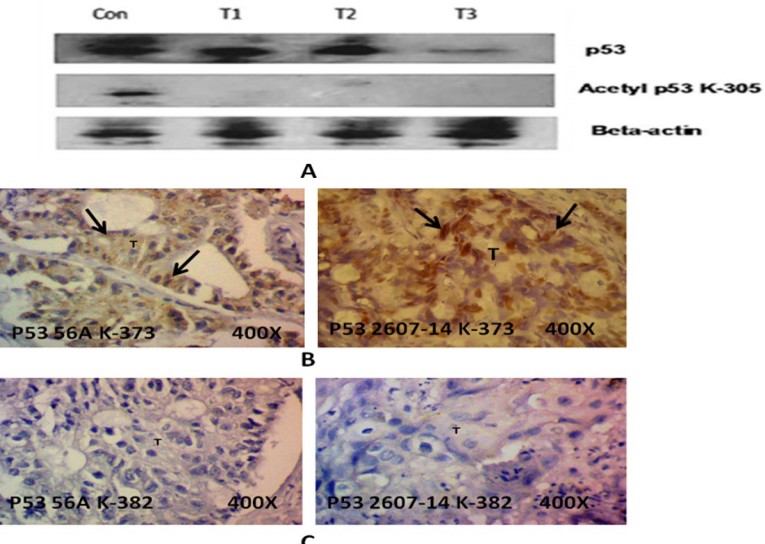

**Fig 6.** (A) Representative western blot showing the absence of p53 acetylation in HPV positive cervical cancer cases compare to non-neoplastic control at site K305. (B) Representative panel of IHC showing the presence of p53 acetylation in HPV positive cervical cancer at site K373. The positive results were observed in very few cases. (C) Representative panel of IHC showing the absence of p53 acetylation in HPV positive cervical cancer at site K382.

## Discussion

In Indian females belonging to both urban as well as rural area, cervical cancer has emerged to be one of the major causes of cancer associated deaths. The poor prognosis, diagnosis and failure to implement primary prevention strategies such as cervical screening and HPV vaccination programs are the major factors involved in mortality due cervical cancer in women in southeast Asia particularly in India [21, 22]. As per the National Centre for Disease Informatics and Research [23] based report, the cancer cervix distribution rate [relative proportion % age (AAR)] in Northeast India ranks amongst the highest prevalent areas or districts in India; making it a major health issue amongst females. Although infection with high risk HPV strains has been underlined as the driving force in cervical cancer pathogenesis [24] along with the data from the present studied cohort, but, the heterogeneity in the presentation and progression of the disease strongly indicates the significance of host factors in the susceptibility and severity of the disease. Given the differences in cervical cancer distribution in different geographical subsets of populations with distinct and different ethnicity [25], the understanding of these key molecular and cellular factors becomes relevant and holds clinical significance.

The inactivation or malfunctioning of p53 a tumor suppressor gene has been found to play a crucial role in the pathogenesis of various cancer which also includes cervical cancer [6, 26]. In our study it was observed that both at mRNA and protein level the expression of P53 was downregulated in cervical cancer cases compared to non-neoplastic control thus indicating its significant role in the malignant transformation and occurrence of cervical cancer. Our findings correlate with several studies which demonstrate that in cervical cancer carcinogenesis the downregulation of p53 levels is thought to be a key factor [26]. The data for p53 downregulation in HPV infected cervical cancer cases with different severity grade also showed a variation with significant downregulation in lower severity grades compare to higher severity grades indicating the association of deregulated P53 expression in the early stage of cervical cancer cases leading to the pathogenesis of the diseases which relates with the study of Liu et al. [27], who suggested that p53 is associated in the development and metastasis of cervical cancer by promoting tumor cell infiltration. Thus, data from our study suggests that p53 downregulation may probably a crucial early event in the pathogenesis of CaCx pathogenesis in HPV infected cases. In the present study the majority of cases were HPV infected so it is suggestive that the viral oncogene i.e E6 might have significant contribution in the malfunction and degradation of p53 protein thus resulting in cervical cancer [28].

Mutations in p53 gene are common in carcinomas and it may result in dysregulation of gene expression of p53 or the disrupted functioning of protein [11, 29]. Apart from p53 expression, the p53 mutation is also associated with the cervical neoplasia. There is a significant correlation between p53 mutation and the change in the expression profile of p53 in both HPV positive and HPV negative cervical cancer cases. Missense mutation is the most common type of mutation which occur in tumor associated with alteration in p53 which lead to single amino acid substitution, also in some cases the frameshift or nonsense mutation can cause the loss of expression of p53 protein [30]. In the present study to evaluate the role of p53 mutation in cervical cancer pathogenesis we studied the mutation in p53 for six different exons (Exon 4 to 9) in the enrolled cases and our data showed the change in exon4 of the p53 gene (rs1042522: Pro72Arg, C/G). The *p53 Arg* genotype (homo or heterozygous state) is known to be targeted by HPV onco-protein E6 and results in deregulated apoptosis [31, 32]. The result obtained for p53 polymorphism for HPV positive cases showed no significant correlation with the risk of developing cervical cancer. Our findings was in consistent with previous report of Yin et al. [33], Tachezy et al. [34], Minaguchi et al. [35] who stated that there was no association between Pro/Pro, Pro/Arg and Arg/Arg residue in the susceptibility of HPV16-postive cervical cancer

whereas it was contrary to the report of Helland et al. [36], Josefsson et al. [37], Lanham et al. [38] who reported that an increased risk of cervical cancer was associated with the Arg 72 TP53 allele.

Further we performed an analysis to find if there was any association between the changes in the expression of p53 with the change in genotype. The data so obtained was interesting which showed that in HPV negative cervical cancer the expression of p53 was upregulated for Arg/arg and Arg/Pro genotype compare to Pro/Pro whereas for in HPV positive cervical cancer the results were completely contrasting showing downregulation in p53 expression for Arg/Arg and Arg/Pro compare to Pro/Pro genotype. The data is therefore suggestive of the prognostic significance of the rs1042522 polymorphism in the studied population, and especially the Arg allele in HPV+ve cases. From the findings we can suggest that the expression pattern base on genotype is associated with HPV status thus indicating that person with HPV 16 infection and homozygous proline residues compare to Arg residues are more susceptible to p53 degradation, though the report is contradictory to the findings of Storey et al. [32] who have reported that persons homozygous for arginine at residue 72 of p53 (p53Arg) are about seven times more prone to invasive cervical cancer than those who carry at least one proline at that position (p53Pro). Our findings are related to another study by Josefsson et al. [37] who reported that p53Arg is not associated with an increased risk of preinvasive or invasive cervical neoplasia; indeed, there is a tendency for p53Arg to be associated with a decreased risk of neoplasia. The observation also highlights the fact that viral oncogene also play a critical role in the progression of cervical carcinoma by resulting in malfunctioning of p53 the tumor suppressor protein.

It has been increasingly realized that epigenetic signatures hold key to upholding the genomic integrity of an individual, which may have detrimental consequences. High risk HPV E6 and E7 oncoproteins have been reported to increase the activity of De novo DNA methyltransferases [39–41]. The risk of developing cancer due to hypermethylation of promoter of p53 has been reported in various cancers such as hepatocellular carcinoma, breast cancer and ovarian cancer [13, 42, 43]. The results from the northeast Indian patient cohort showed that the p53 promoter methylation profile were heterogeneous, and also suggested that the hyper-methylation status was associated with the lower severity grade (stage II) in comparison to severe stages (stage IV). This is contrary to earlier published reports from North Indian population [44] where it was reported that promoter hypermethylation was mainly observed in the serum samples in the higher stages and very rarely in the lower stages. Acetylation is responsible to deliver stability to p53 and enhance its function while the hypo-acetylation of gene is known to represses expression [10, 15]. Eight major acetylation sites in p53 acetylation sites has been documented [17] but few of them like K305, 373 and 382 are of major relevance because of its association with the transcriptional activity of p53 and its interaction with key signalling regulators like p300 and HDAC SirT1 [45]. The acetylated p53 at K373/K382 are reported to enhance the half-life of p53 and decrease its ubiquitination. The acetylation of p53 is also crucial for its binding to downstream target [45, 46]. In the current study we corroborate the previously reported data of the expression of p53, the loss of acetylation at K305 and K382 can be associated with the down regulation of p53 in northeast Indian population. The data is of clinical relevance as the efficiency of several chemo- and radiotherapies has been suggested to be principally p53 tumor suppressor gene dependent in many cases [47] which in turn has been proven to be regulated by high-risk HPV E6 protein expression [48, 49] or dependent of histone deacetylases (HDAC) like SirT1 expression dependent [50]. Indeed the significance of inhibition of HDACs as promising target in cancer therapeutics has been increasingly realized [51]. The results on the p53 acetylation changes and resulting expression in CaCx cases is also important because p53 acetylation in response to virus infection has been documented as an

indispensable event for the transcriptional activation of p53-dependent genes in reaction to viral infection and the successive control of virus replication [52] which is mainly mediated by activation of the interferon pathway [53].

## Conclusion

Although the number of samples was limited in our study, the present study for the first time highlights the role of deregulated p53 in the pathogenesis of cervical cancer in the ethnically distinct population of Northeast India. Our study indicates that changes in p53 profile such as p53 mutation, down regulation in expression along with the changes in p53 epigenetic profile is associated with the pathogenesis of cervical cancer, though substantial amount of data with large cohort study is necessary to prove the findings. We also observed the significant deregulation in p53 in early stage of cancer thus indicating that, significant role of p53 as tumor suppressor gene in the cervical cancer may establish it as possible biomarker in the future and can play vital role in diagnosis and therapeutic targeting.

## Supporting information

**S1 Table. Details of p53 exon4 polymorphism based difference in genotype analysis.**
(DOCX)

**S1 File.**
(DOCX)

**S2 File.**
(DOCX)

**S3 File.**
(PDF)

**S4 File.**
(PDF)

**S5 File.**
(PDF)

**S6 File.**
(ZIP)

## Acknowledgments

The authors acknowledge Department of Biotechnology (DBT), Govt. of India, for financially supporting the study. The authors also acknowledge Dr. Mukherjee, New Delhi, for the immunohistochemistry analysis.

## Author Contributions

**Conceptualization:** Mohammad Aasif Khan, Diptika Tiwari, Purabi Deka Bose, Sujoy Bose, Syed Akhtar Husain.

**Data curation:** Mohammad Aasif Khan, Diptika Tiwari, Purabi Deka Bose, Sujoy Bose, Syed Akhtar Husain.

**Formal analysis:** Diptika Tiwari, Sadaf, Saad Mustafa, Purabi Deka Bose, Sujoy Bose, Syed Akhtar Husain.

**Funding acquisition:** Purabi Deka Bose, Sujoy Bose, Syed Akhtar Husain.

**Investigation:** Mohammad Aasif Khan, Purabi Deka Bose, Sujoy Bose, Syed Akhtar Husain.

**Methodology:** Mohammad Aasif Khan, Sadaf, Saad Mustafa.

**Project administration:** Purabi Deka Bose, Sujoy Bose, Syed Akhtar Husain.

**Resources:** Chandana Ray Das.

**Software:** Anita Dongre.

**Supervision:** Syed Akhtar Husain.

**Validation:** Mohammad Aasif Khan, Diptika Tiwari, Sadaf, Saad Mustafa, Purabi Deka Bose, Syed Akhtar Husain.

**Visualization:** Mohammad Aasif Khan, Diptika Tiwari, Saad Mustafa, Purabi Deka Bose, Syed Akhtar Husain.

**Writing – original draft:** Mohammad Aasif Khan, Saad Mustafa, Purabi Deka Bose, Sujoy Bose, Syed Akhtar Husain.

**Writing – review & editing:** Mohammad Aasif Khan, Sheersh Massey, Syed Akhtar Husain.

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
