## [Decision Letter · Decision Letter 0]

28 Jan 2020

PONE-D-19-32000

Screening the p53 Connection of Cervical Cancer Pathogenesis Involving North-East Indian Patients.

PLOS ONE

Dear Dr. Husain,

Thank you for submitting your manuscript to PLOS ONE. After careful consideration, we feel that it has merit but does not fully meet PLOS ONE’s publication criteria as it currently stands. Therefore, we invite you to submit a revised version of the manuscript that addresses the points raised during the review process.

ACADEMIC EDITOR: Brief methodologies needed for the conducted experiments and the image quality should be improvised.

We would appreciate receiving your revised manuscript by 27th Feb 2020. To enhance the reproducibility of your results, we recommend that if applicable you deposit your laboratory protocols in protocols.io, where a protocol can be assigned its own identifier (DOI) such that it can be cited independently in the future. For instructions see: http://journals.plos.org/plosone/s/submission-guidelines#loc-laboratory-protocols

We look forward to receiving your revised manuscript.

Kind regards,

Kalimuthusamy Natarajaseenivasan

Academic Editor

PLOS ONE

Journal Requirements:

3. Please include your tables as part of your main manuscript and remove the individual files. Please note that supplementary tables (should remain/ be uploaded) as separate "supporting information" files.

6. We noticed you have some minor occurrence(s) of overlapping text with the following previous publication(s), which needs to be addressed:

https://doi.org/10.3390/cancers7010030

https://doi.org/10.1038/25040

https://doi.org/10.4161/cc.10.21.17899

In your revision ensure you cite all your sources (including your own works), and quote or rephrase any duplicated text outside the Methods section. Further consideration is dependent on these concerns being addressed.

7. To comply with PLOS ONE submission guidelines please deposit your sequencing data in a publicly available repository (you can find a list of repositories in the link here : https://journals.plos.org/plosone/s/data-availability#loc-recommended-repositories).

8. We suggest you thoroughly copyedit your manuscript for language usage, spelling, and grammar. If you do not know anyone who can help you do this, you may wish to consider employing a professional scientific editing service.  

9. Please clarify why ethical approval was not obtained from the ethical committee of Gauhati Medical College and Hospital, where patient samples were collected from

10. Please provide additional details regarding participant consent. In the ethics statement in the Methods and online submission information, please ensure that you have specified what type of consent you obtained (for instance, written or verbal). If your study included minors under age 18, state whether you obtained consent from parents or guardians. If the need for consent was waived by the ethics committee, please include this information.

Reviewers' comments:

Reviewer's Responses to Questions

**Comments to the Author**

1. Is the manuscript technically sound, and do the data support the conclusions?

Reviewer #1: Yes

Reviewer #2: Partly

2. Has the statistical analysis been performed appropriately and rigorously? 

Reviewer #1: Yes

Reviewer #2: No

3. Have the authors made all data underlying the findings in their manuscript fully available?

Reviewer #1: Yes

Reviewer #2: Yes

4. Is the manuscript presented in an intelligible fashion and written in standard English?

Reviewer #1: No

Reviewer #2: Yes

5. Review Comments to the Author

Reviewer #1: In the manuscript entitled “Screening the p53 Connection of Cervical Cancer Pathogenesis Involving North-East Indian Patients” the authors have discussed the mutation and expression aspects of p53 in cervical cancer pathogenesis

Though the study is relevant as well as conceptually and methodologically sound, however suffers from writing and presentation issues, therefore need major revision.

Some of the major and minor points are as follows:

Abstract:

Background: “Cervical cancer (CaCx) is a global issue” how?? Needs clarity

Human papilloma virus should read: Human papillomavirus

Blood was collected but how was it utilized is not mentioned??

Keywords: CaCx should be deleted

Introduction:

It is a leading cause of cancer related mortality in Indian women of both rural and urban areas??

Well breast cancer in India leads! Latest reference/ GLOBOCAN report should be referred

First Para: last sentence “biomarker” to be deleted

Ref. no. 8 and 9 do not relate to the statements made.

The statement “The data in the role of dysregulated p53 in CaCx pathogenesis has been suggested, but differences exists in the data documented from different geographical niches” needs elaboration and citation.

First page: last sentence: Tp53 should read: TP53

The statement “Genetic alterations in p53 gene have seldom been reported in carcinogenesis event [11]” is baffling.

The authors should explain the ration behind this statement or it should be deleted. There is no dearth of reports on the genetic alterations of p53 in cancer.

Last paragraph: Mizoram study??? Cancer of other etiology?? Which cancer?? Reference is missing?? The paragraph is ambiguous

Materials and methods

Enrolment of Patient and sample collection: Authors have discussed the processing of the samples. The title should be modified accordingly.

Statement on ethical approval is missing.

The ethnicity of the population needs clarity, and how the present population is different from the other population of their country.

Did the patient samples (blood and biopsies) were collected by Gynaecologist or by the non-clinicians under the supervision of Gynaecologist. The statement needs to be reframed.

Why blood was collected??

(for RNA bases study) ??

The following statement should be the part of results and not materials and methods

“and based on the screening report, 84% (N=72) of cervical cancer cases were found to be HPV positive.”

TP53 expression analysis

Sybr green should be SYBR Green

IHC: scoring system not mentioned

TP53 Epigenetic profiling:

the paired HPV positive cases (i.e. cases with both cancerous and noncancerous

region) specifically for data specificity.

Above statement is not clear: what about HPV negative cancer cases

At many places Kit nomenclature is incomplete, country of origin is missing

Several sites of TP53 known to be acetylated e. g, C-terminal K370, K372, K373, K381, K382, as well as K120, K164, K305 etc. However, rational of selection K305, K373 and K382 should be discussed.

P53 Acetylation Study

For which type Ab, WB and IHC was used?? Details should be provided here

Results

Demographical and clinical profile of the enrolled patients

Clinical profile is missing.

No association of the results with different stages of CaCx has been made and discussed

Percentage missing in HPV results

How stage III can be common to both low and high stages of cervical cancer: “lower severity stage (≤stage III) compared to higher stages of CaCx (≥stage III)”

“To check the role of altered p53 gene in the susceptibility to CaCx the mutation study for six

different exons (exon4�9) was done by PCR direct sequencing method (Fig 4A)” lacks clarity and should be reframed

“The analysis result from the sequencing showed changes in exon4 of the p53 gene only in the affected region of the studied cases” ??

What is “the affected region of the studied cases”

The following statement should be shifted to the discussion part. “This data suggests that p53 downreglation may probably a crucial early event in the pathogenesis of CaCx pathogenesis in HPV infected cases“

Following statement is the part of results and must be shortened and shifted to the discussion

“Both the two isoforms of p53 due to polymorphism at codon 72 differ in biochemical and

biological properties, especially in reference to its property of inducing apoptosis [19].

Sporadic report also is suggestive of the significance of HPV onco-protein E6 in targeted

degradation of Arg72 p53, thereby altering cellular apoptosis and predisposing individuals to

HPV-related cervical cancers [20]. Secondly, literature suggests that Arg/Pro germline

heterozygotes is retained in squamous cell carcinomas and is more potent in neutralizing p73-

induced apoptosis and also cooperates with other oncogenes to transform cells [21].”

Both the two isoforms of p53??

“The data is therefore suggestive of the prognostic significance of the rs1042522 polymorphism in the studied population, and especially the Arg allele in

HPV+ve cases”

Must go to discussion

When the cases where changes in p53 promoter methylation changes observed were considered ??? Needs clarity

Acetylation also plays a key role in p53 protein stabilization and transcriptional regulations.

Key acetylation sites which is associated with p53 transcript stability which inturn p53

protein expression K305, 373 and 382 was analyzed for differential expression with p53-

acetylation specific antibodies by either western blot analysis (Ac-305) or by immunohistochemistry (Ac-372 and Ac-382).

The above statement cannot be part of the results: half goes to MM and rest to discussion

Discussion:

Josefesson A et al. 1998 should read: Josefesson et al. (1998)

There are many such incorrect citations throughout the manuscript

“In Indian females belonging to both urban as well as rural area, cervical cancer has emerged

to be major cause of cancer associated deaths” OR CaBr.

Following statement is too long and lacks clarity

“As per the National Centre for Disease Informatics and Research [24] based report on Cancer Burden in North Eastern States of India, the prevalence of cancer cervix in northeast India stands at 12.3 (10.1) [relative proportion %age (AAR)]; and more importantly, several districts of Arunachal Pradesh, Mizoram, Assam and Nagaland have high to very high distribution of the disease (rate per 100,000) and ranks amongst the highest prevalent areas or districts in India; making it a major health issue amongst females”

“Although infection with high risk HPV strains has been underlined as the driving force in cervical cancer pathogenesis [25]. along with the data from the present studied cohort”

What about high-risk HPV stains’ data in the present cohort. No mention has been made in the present study. Authors should provide data about high-risk HPVs analysed and their association.

If northeast states are Arunachal Pradesh, Mizoram, Assam and Nagaland of India then from which state patients were recruited. Rational for selecting NE patients is not mentioed?? How this population ethinically differs from the other part of the country. Future comparative studies will other ethnic groups will be required to get a clear picture. This needs to be discussed.

“Given the differences in cervical cancer distribution in different geographical

subsets of populations with distinct and different ethnicity, the understanding of these key

molecular and cellular factors becomes relevant and holds clinical significance”

Reference is missing

“Our findings correlate with several studies which demonstrates that in cervical cancer

carcinogenesis the down-regulation of p53 levels is thought to be a key factor [26]”

In the above statement authors says “several studies”, however only one reference has been cited and this reference is not a review article.

What about p53 expression in HPV negative CaCx cases: No indication in results and discussion

As cancer is known to be a multigene effect, how authors can signify the importance of only p53 and not the other genes. This needs to be argued.

Table 1 . p53 should be italicised

Table 2: Number of cases [%age].

% not calculated

Table 2 and 3: CaCx should read: cervical cancer

At many places figure legends are incomplete., e.g., Fig 4A

What is Con?? Fig A, B,C??

The authors should avoid using “to check” frequencly. Instead, relevant synonyms should be used.

There are innumerable full stops in-between the statements.

e.g., Inactivation of p53 has been reported to be due to hyper-methylation of the promoter region [14]. and it has been associated with human neoplasia [15].

A review of English by a native English speaker or assistance from a professional English editing services is strongly recommended.

Reviewer #2: Comments:

The authors Khan et al. tried to explore the p53 connection of cervical cancer pathogenesis in North-east Indian patients by studying the complete p53 profiling including differential mRNA expression, immunohistochemistry, mutational status of p53 and also epigenetic profiling. In spite of a large number of publications on this subject, this manuscript carries importance as there is still a huge cervical cancer burden on the North-east Indian women which encourages the scientific community and researcher to continue the studies on cervical cancer. This manuscript has tried to find the significant role of p53 which is one of the most important tumor suppressor genes in CaCx patients of North-east Indian women.

Going through the manuscript, I found several major concerns to be addressed by authors -

1. In the title the word ‘Screening’ may be replaced by words like ‘Evaluating’ or ‘Exploring’.

2. According to the statement made by the authors in the Introduction section that “The difference in the clinical presentation, the progression of the disease as well as response to treatment differs amongst individuals both with and without underlying HPV infection”, therefore both HPV positive and HPV negative CaCx cases carry equal significance for exploration of the mRNA expression for p53, however mRNA expression has only been studied for HPV positive CaCx cases by the authors. So what is the inference for HPV negative CaCx cases having downregulation of p53 mRNA expression? Comment.

3. In the Materials and Method section please mention what experiments were conducted with the blood samples collected from the patients.

4. Generally, 10% formalin is used for tissue fixation which contains 4% formaldehyde. So kindly comment on the utility of 4% formalin in tissue fixation.

5. Please provide brief protocol for each of the experiments conducted as per requirement of the journal. Also provide the details of the primary antibody against p53 used for IHC.

6. More rigorous statistical data needs to be furnished.

7. The distribution of the age group mentioned in the results section does not match with the table provided by the authors (Table 2).

8. The half life of the wild type p53 is very short and thus difficult to detect by IHC so the authors are requested to clarify whether the p53 detected by IHC for the study of differential p53 protein expression is wild type or mutated p53.

9. IHC images provided are low in resolution and lacking professional quality for publication. It is advised to include high resolution images containing arrows to mark the area of interest and scaling of the images. Moreover, please recheck the magnification mentioned in the images as it is written 400X and also the figure legends are not clear enough. Western blot images are not satisfactory.

10. The authors should comment on the limitation of the study as definite conclusion cannot be drawn from such small population of cases.

6. PLOS authors have the option to publish the peer review history of their article (what does this mean?). If published, this will include your full peer review and any attached files.

Reviewer #1: No

Reviewer #2: No

---

## [Author Response · Author response to Decision Letter 0]

26 Feb 2020

Comments to the Author

1. Is the manuscript technically sound, and do the data support the conclusions?

Reviewer #1: Yes

Reviewer #2: Partly

2. Has the statistical analysis been performed appropriately and rigorously?

Reviewer #1: Yes

Reviewer #2: No

3. Have the authors made all data underlying the findings in their manuscript fully available?

Reviewer #1: Yes

Reviewer #2: Yes

4. Is the manuscript presented in an intelligible fashion and written in standard English?

Reviewer #1: No

Reviewer #2: Yes

5. Review Comments to the Author

Reviewer #1 Comments:

 In the manuscript entitled “Screening the p53 Connection of Cervical Cancer Pathogenesis Involving North- East Indian Patients” the authors have discussed the mutation and expression aspects of p53 in cervical cancer pathogenesis

Though the study is relevant as well as conceptually and methodologically sound, however suffers from writing and presentation issues, therefore need major revision.

Some of the major and minor points are as follows:

Abstract:

Background: “Cervical cancer (CaCx) is a global issue” how?? Needs clarity

 Response: As instructed the clarification is being made and the same has been highlighted in 

 the text in blue.

Human papilloma virus should read: Human papillomavirus

Response: The necessary correction has been made in the text and highlighted in blue.

Blood was collected but how was it utilized is not mentioned??

Response: The blood samples were collected for screening out any other infective pathology like HIV, HBV etc which were used as exclusion criteria in our study. 

Keywords: CaCx should be deleted

 Response: CaCx from the keyword has been deleted.

Introduction: 

It is a leading cause of cancer related mortality in Indian women of both rural and urban areas?? Well breast cancer in India leads! Latest reference/ GLOBOCAB report should be referred 

Response: The sentence has been mentioned as “a leading cause” and not “the leading cause”. As per recent GLOBOCON data cervical cancer in India is the second most ranked cancer in women (16.5%) accounting for 60,000 death in 2018. 

First Para: last sentence “biomarker” to be deleted

Response: As instructed, the word “biomarker” has been deleted from the text.

Ref. no. 8 and 9 do not relate to the statements made. 

Response: The reference has been corrected and incorporated as per the statement and has been highlighted in blue in the text.

The statement “The data in the role of dysregulated p53 in CaCx pathogenesis has been suggested, but differences exists in the data documented from different geographical niches” needs elaboration and citation.

Response: As instructed the citation for the above statement has been provided and the same has been highlighted in blue

First page: last sentence: Tp53 should read: TP53

Response: As instructed the necessary correction has been made and the same has been highlighted in the text in blue

The statement “Genetic alterations in p53 gene have seldom been reported in carcinogenesis event [11]” is baffling. The authors should explain the ration behind this statement, or it should be deleted. There is no dearth of reports on the genetic alterations of p53 in cancer.

Response: The line as suggested has been deleted and hence corrected.

Last paragraph: Mizoram study??? Cancer of other etiology?? Which cancer?? Reference is missing?? The paragraph is ambiguous.

Response: As instructed the references pertinent to the association of p53 mutation with different cancer aetiologias has been enlisted and incorporated.

Materials and methods

Enrolment of Patient and sample collection: Authors have discussed the processing of the samples. The title should be modified accordingly. 

Response: As instructed the separate title for processing of sample has been incorporated and the same has been highlighted in blue in the text.

Statement on ethical approval is missing.

Response: Statement stating “The study was permitted by the institutional ethics committees of all the participating institutes” has been incorporated in the text and highlighted in blue.

The ethnicity of the population needs clarity, and how the present population is different from the other population of their country.

Response: As instructed the clarity for the ethnicity of the northeast Indian population has been incorporated in the text and the same has been highlighted in the text in blue.

Did the patient samples (blood and biopsies) were collected by Gynecologist or by the non-clinicians under the supervision of Gynecologist. The statement needs to be reframed.

Response: As instructed the statement has been reframed and the collection of samples by Gynecologist has been incorporated in the text and the same has been highlighted in blue.

Why blood was collected?? 

(for RNA bases study)??

Response: : The blood samples were collected for screening out any other infective pathology like HIV, HBV etc which were used as exclusion criteria in our study and the statement stating the reason has been incorporated in the text in materials and methods section and same has been highlighted in blue.

The following statement should be the part of results and not materials and methods

 “and based on the screening report, 84% (N=72) of cervical cancer cases were found to be 

 HPV positive.”

 Response: As instructed the sentence has been removed from the materials and methods

 section and has been incorporated in the result section and the same have been highlighted in 

 blue in the text

TP53 expression analysis

Sybr green should be SYBR Green

Response: As instructed the word Sybr green has been changed to SYBR Green and the same has been highlighted in text in blue.

IHC: scoring system not mentioned

Response: The scoring system for IHC has been incorporated in the text in materials and methods section and the same has been highlighted in blue.

TP53 Epigenetic profiling:

the paired HPV positive cases (i.e. cases with both cancerous and noncancerous region)

 specifically, for data specificity.

 Above statement is not clear: what about HPV negative cancer cases

 Response: Since the number of HPV negative cancer cases were very less in number (n=13 

 out of 85 cases) compared to HPV positive cases, therefore it would have not been possible 

 to generate statistically sound relevant data with less number of cases so it was excluded 

 from the present study.

At many places Kit nomenclature is incomplete, country of origin is missing

Response: As instructed the missing nomenclature for kit has been incorporated in the text and has been highlighted in the text.

Several sites of TP53 known to be acetylated e. g, C-terminal K370, K372, K373, K381, K382, as well as K120, K164, K305 etc. However, rational of selection K305, K373 and K382 should be discussed. 

Response: We are aware that several sites of TP53 are known to be acetylated but in our study we picked up three site i.e. K373, K381 and K305 because of its relevancy with the transcriptional activity and stability of P53 which is the main focus of our study, and the same has been discussed in our manuscript as “K305, 373 and 382 are of major relevance because of its association with the transcriptional activity of p53 and its interaction with key signaling regulators like p300 and HDAC SirT1. The acetylated p53 at K373/K382 are reported to enhance the half-life of p53 and decrease its ubiquitination. The acetylation of p53 is also crucial for its binding to downstream target.” The other acetylation sites are found to be associated with cellular apoptosis which is not under the preview of our study and so we excluded those sites. 

P53 Acetylation Study

For which type Ab, WB and IHC was used?? Details should be provided here

Response: As instructed the details about the type of antibodies used for WB and IHC for acetylation study has been incorporated in the text in the section- P53 acetylation study, and the same has been highlighted in the text in blue.

Results

Demographical and clinical profile of the enrolled patients

Clinical profile is missing.

Response: The demographical and clinical profile of CaCx cases have been stated in Table 2 in the text. Additionally we have incorporated the gravida status of enrolled cases.

No association of the results with different stages of CaCx has been made and discussed

Response: It has been already mentioned in the text as follows “Since majority of the cases had an underlying HPV infection (84%), therefore the differences in p53 mRNA expression in different stages of CaCx was further compared. The downregulation of p53 mRNA expression was not uniform in CaCx cases of different severity grades. The p53 mRNA expression was downregulated in lower severity stage (stage IIA and Stage IIB) compared to higher stages of CaCx (≥stage III) (Fig 2A). The expression of p53 was significantly downregulated in stage II compared to stage III (p=0.026) and stage IV (p=0.049) also p53 expression was downregulated in stage III compared to stage IV (Fig 2B).”

Percentage missing in HPV results

Response: As instructed the percentage for all the HPV results has been incorporated in the table 2 of the manuscript and the same has been highlighted in the text in blue.

How stage III can be common to both low and high stages of cervical cancer: “lower severity stage (≤stage III) compared to higher stages of CaCx (≥stage III)”

Response: Only stage IIA and IIB has been considered as lower grade severity and stage III and above were considered as higher grade of severity and the same has been rectified in text and highlighted in blue.

“To check the role of altered p53 gene in the susceptibility to CaCx the mutation study for six different exons (exon4�9) was done by PCR direct sequencing method (Fig 4A)” lacks clarity and should be reframed.

Response: As instructed the sentence has been reframed and also the figure legend for Fig4A has been modified and incorporated in the text and the changes made have been highlighted in blue in the text.

“The analysis result from the sequencing showed changes in exon4 of the p53 gene only in the affected region of the studied cases”??

Response: There was a typographical error hence the sentence has been corrected and incorporated as “The analysis result from the sequencing showed changes in only exon4 of the p53 gene in all the enrolled cases” and the correction has been highlighted in text in blue.

What is “the affected region of the studied cases”

Response: It was typographical error, so it has been removed from the text and the sentence was reframed accordingly. We apologize for the mistake.

The following statement should be shifted to the discussion part. “This data suggests that p53 downregulation may probably a crucial early event in the pathogenesis of CaCx pathogenesis in HPV infected cases”

Response: As instructed the above-mentioned statement has been shifted to discussion part and the same has been highlighted in text in blue.

Following statement is the part of results and must be shortened and shifted to the discussion “Both the two isoforms of p53 due to polymorphism at codon 72 differ in biochemical and biological properties, especially in reference to its property of inducing apoptosis [19]. Sporadic report also is suggestive of the significance of HPV onco-protein E6 in targeted degradation of Arg72 p53, thereby altering cellular apoptosis and predisposing individuals to HPV-related cervical cancers [20]. Secondly, literature suggests that Arg/Pro germline heterozygotes is retained in squamous cell carcinomas and is more potent in neutralizing p73- induced apoptosis and also cooperates with other oncogenes to transform cells [21].”

Response: The above mentioned statement was shortened and as instructed was incorporated in the discussion section as “The p53 Arg genotype (homo or heterozygous state) is known to be targeted by HPV onco-protein E6 and results in deregulated apoptosis [19, 20, 21]” .and the same has been highlighted in text in blue.

Both the two isoforms of p53?? 

Response: The words ”both the two isoforms of p53 ( Arg/Arg and Arg/Pro) from the sentence has been removed and the sentence was reframed as “The p53 Arg genotype (homo or heterozygous state) is known to be targeted by HPV onco-protein E6 and results in deregulated apoptosis [19, 20, 21]” in discussion section.

“The data is therefore suggestive of the prognostic significance of the rs1042522 polymorphism in the studied population, and especially the Arg allele in HPV+ve cases” Must go to discussion

 Response: As instructed the above statement has been incorporated in discussion section and the same has been highlighted in blue in the text.

When the cases where changes in p53 promoter methylation changes observed were considered??? Needs clarity

Response: As instructed the sentence has been reframed and corrected as “When we considered the cases where changes in p53 promoter methylation was observed, the differences in methylation profile indicated that p53 promoter hyper-methylation was significantly associated lower severity grade (stage IIA and IIB)” and has been incorporated in the text and same has been highlighted in blue oion the text

Acetylation also plays a key role in p53 protein stabilization and transcriptional regulations. Key acetylation sites which is associated with p53 transcript stability which inturn p53 protein expression K305, 373 and 382 was analyzed for differential expression with p53- acetylation specific antibodies by either western blot analysis (Ac-305) or by immunohistochemistry (Ac-372 and Ac-382). 

The above statement cannot be part of the results: half goes to MM and rest to discussion Response: As instructed the above statement has been removed from the result section section and has been incorporated in materials and methods section. The correction made has been highlighted in blue in the text.

Discussion:

Josefesson A et al. 1998 should read: Josefesson et al. (1998)

There are many such incorrect citations throughout the manuscript

Response: As instructed all the necessary correction for citation has been made in the text and the same has been highlighted in blue.

“In Indian females belonging to both urban as well as rural area, cervical cancer has emerged to be major cause of cancer associated deaths” OR CaBr.

Response: As instructed the sentence has been reframed as “In Indian females belonging to both urban as well as rural area, cervical cancer has emerged to be one of the major cause of cancer associated deaths” and has been incorporated in the text.

Following statement is too long and lacks clarity

“As per the National Centre for Disease Informatics and Research [24] based report on Cancer Burden in North Eastern States of India, the prevalence of cancer cervix in northeast India stands at 12.3 (10.1) [relative proportion %age (AAR)]; and more importantly, several districts of Arunachal Pradesh, Mizoram, Assam and Nagaland have high to very high distribution of the disease (rate per 100,000) and ranks amongst the highest prevalent areas or districts in India; making it a major health issue amongst females”

Response: As per the instruction the following sentence has been shortened and reframed and has been incorporated in text as “As per the National Centre for Disease Informatics and Research [24] based report, the cancer cervix distribution rate [relative proportion %age (AAR)] in Northeast India ranks amongst the highest prevalent areas or districts in India; making it a major health issue amongst females.” and the same has been highlighted in blue.

“Although infection with high risk HPV strains has been underlined as the driving force in cervical cancer pathogenesis [25]. along with the data from the present studied cohort”. What about high-risk HPV stains’ data in the present cohort. No mention has been made in the present study. Authors should provide data about high-risk HPVs analysed and their association.

Response: HPV 16 is also one of the high-risk HPV and is the most predominant genotype in the Brahmaputra valley (Northeat India). The prevalence for dominant HPV16 genotype has also been reported previously by Das et al 2013 as well as in our study. So, in the present study our main focus was HPV16 infected cervical cancer cases which was predominant in our collected cervical cancer cases.

If northeast states are Arunachal Pradesh, Mizoram, Assam and Nagaland of India then from which state patients were recruited. Rational for selecting NE patients is not mentioned?? How this population ethnically differs from the other part of the country. Future comparative studies will other ethnic groups will be required to get a clear picture. This needs to be discussed.

Response: The samples were collected from Gauhati Medical College and Hospital, Assam which is a hub for the patients from different Northeastern India states such as Meghalaya, Tripura, Manipur, Nagaland etc belonging to different ethnicity group basically the tribal dominant and those who differ from the other parts of India. As there is no p53 based study in context to cervical cancer from this region, so considering the geographical location and ethnicity of population difference in Northeastern region we undertook the study on cervical cancer pathogenesis in Northeast Indian population. The explanation for the difference in ethnic group has also been provided in the response in Materials and methods section for Enrolment of Patient and Sample Collection.

“Given the differences in cervical cancer distribution in different geographical subsets of populations with distinct and different ethnicity, the understanding of these key molecular and cellular factors becomes relevant and holds clinical significance” Reference is missing

Response: As instructed the reference for the above statement has been incorporated in the text and the same has been highlighted in blue.

“Our findings correlate with several studies which demonstrates that in cervical cancer carcinogenesis the down-regulation of p53 levels is thought to be a key factor [26]”.

In the above statement authors says, “several studies”, however only one reference has been cited and this reference is not a review article.

Response: As instructed, for the above statement the reference of a review article have been incorporated in the text and the same has been highlighted in blue.

What about p53 expression in HPV negative CaCx cases: No indication in results and discussion.

Response: We did mention in the result about P53 expression for HPV negative cases but as mentioned in previous response in material and method section that due to the less number of HPV negative cancer cases (n=13 out of 85 cases ) compared to HPV positive cases, it would have not been possible to generate statistically sound relevant data and so it was excluded from the present study. 

As cancer is known to be a multigene effect, how authors can signify the importance of only p53 and not the other genes. This needs to be argued.

Response: The p53 protein is recognized as the most frequently inactivated tumor suppressors in human cancers as its has been documented in affecting many important cellular processes including proliferation, DNA repair, programmed cell death (apoptosis), autophagy, metabolism, and cell migration. The p53 protein is the centre of many crosstalk’s with critical signaling transducers thereby effecting cell cycle, apoptosis and other hall marks of cancer. The p53 has been stated to be an early hit during cancer pathogenesis of multiple etiologies. Although cancer is a multi-step event, but the role of p53 in cellular homeostasis augments its significance to be studied in cancer contexts includingHPV16 linked cervical cancer as done in the present study. 

Table 1 . p53 should be italicised 

Response: As instructed the necessary correction has been made and is highlighted in text in blue.

Table 2: Number of cases [%age]. 

% not calculated 

Response: As instructed % has been calculated and incorporated in the Table 2 and the same has been highlighted in blue in the table 2.

Table 2 and 3: CaCx should read: cervical cancer

Response: As instructed the word CaCx has been replaced with cervical cancer in both Table 2 and 3 and the same has been highlighted in blue. 

At many places figure legends are incomplete., e.g., Fig 4A

Response: As instructed the figure legends have been elaborated and rectified accordingly and the same has been highlighted in the text in blue.

What is Con?? Fig A, B, C??

Response: In the figure Con represents the non-neoplastic control samples and as instructed in previous query the same has been incorporated in the figure legend. Also, the Figure ^ has been modified and proper designation as A, B, C has been incorporated. The modification has been highlighted in the text as blue.

The authors should avoid using “to check” and instead relevant synonyms should be used. 

 Response: As instructed we have removed the use of “to check” and instead have used 

 relevant synonyms in the text and the same has been highlighted in blue.

There are innumerable full stops in the text in-between the statements.

 e.g., Inactivation of p53 has been reported to be due to hyper-methylation of the promoter 

 region [14]. and it has been associated with human neoplasia [15]. 

 Response: As instructed we have checked the manuscript thoroughly and have remove 

 innumerable full stops in the text in-between the statements.

 A review of English by a native English speaker or assistance from profession English 

 editing services is strongly recommended.

Reviewer #2: Comments:

The authors Khan et al. tried to explore the p53 connection of cervical cancer pathogenesis in North-east Indian patients by studying the complete p53 profiling including differential mRNA expression, immunohistochemistry, mutational status of p53 and also epigenetic profiling. In spite of a large number of publications on this subject, this manuscript carries importance as there is still a huge cervical cancer burden on the North-east Indian women which encourages the scientific community and researcher to continue the studies on cervical cancer. This manuscript has tried to find the significant role of p53 which is one of the most important tumor suppressor genes in CaCx patients of North-east Indian women.

Going through the manuscript, I found several major concerns to be addressed by authors -

1. In the title the word ‘Screening’ may be replaced by words like ‘Evaluating’ or ‘Exploring’.

 Response: As suggested we have replaced the word “Screening” by word “Exploring” 

 and the same has been highlighted in the text in Blue.

2. According to the statement made by the authors in the Introduction section that “The difference in the clinical presentation, the progression of the disease as well as response to treatment differs amongst individuals both with and without underlying HPV infection”, therefore both HPV positive and HPV negative CaCx cases carry equal significance for exploration of the mRNA expression for p53, however mRNA expression has only been studied for HPV positive CaCx cases by the authors. So, what is the inference for HPV negative CaCx cases having downregulation of p53 mRNA expression? Comment.

Response: The similar query were raise by the reviewer1, and in the response we have stated “We did mention in the result about P53 expression for HPV negative cases but as mentioned in previous response in material and method section that due to Since the number of HPV negative cancer cases (n=13 out of 85 cases ) compared to HPV positive cases, it would have not been possible to generate statistically sound relevant data with less number of cases so it was excluded from the present study.” So, our response for the above query will be same as previous one.

3. In the Materials and Method section please mention what experiments were conducted with the blood samples collected from the patients.

Response: As instructed we have mentioned the use of blood sample in our study, in the material and method section and the same has been highlighted in blue in the text.

4. Generally, 10% formalin is used for tissue fixation which contains 4% formaldehyde. So kindly comment on the utility of 4% formalin in tissue fixation.

Response: The tissues were fixed in 10%formalin. There was a typographical error and the error has been corrected in the text as 10% formalin instead of 4% formalin and the same has been highlighted in blue in processing of sample section. We apologize for the mistake.

5. Please provide brief protocol for each of the experiments conducted as per requirement of the journal. Also provide the details of the primary antibody against p53 used for IHC.

Response: As instructed we have provided the brief protocol for each of the experiment at the best possible way, as per the requirement of the journal. We have also provided the details of the primary antibody against p53 used for IHC. The correction has been highlighted in text in blue.

6. More rigorous statistical data needs to be furnished.

Response: The data has been presented in a best possible manner from our end, and the same was acceptable at another Reviewer’s end. If any specific changes are required then it will be easier at our end to rectify our limitations. 

7. The distribution of the age group mentioned in the results section does not match with the table provided by the authors (Table 2).

Response: As instructed the distribution of age group have been corrected in the text and the same has been highlighted in blue. Our apologies for the mistake at our end.

8. The half life of the wild type p53 is very short and thus difficult to detect by IHC so the authors are requested to clarify whether the p53 detected by IHC for the study of differential p53 protein expression is wild type or mutated p53.

Response: Although the half life of p53 is noted to be short under normal conditions, but the role of p53 signifies its expression in altered cellular stress conditions, including those under viral assaults. The expression positivity was noticed in both wildtype and mutated p53 cases. The antibody used i.e. ab26 (Abcam, UK) is anti-p53 monoclonal antibody capable of recognizing both mutant forms and wild-type human p53.

9. IHC images provided are low in resolution and lacking professional quality for publication. It is advised to include high resolution images containing arrows to mark the area of interest and scaling of the images. Moreover, please recheck the magnification mentioned in the images as it is written 400X and also the figure legends are not clear enough. Western blot images are not satisfactory.

Response: As instructed we have incorporated the better resolution images with the incorporation of arrow marks. We have rectified the figure legends and all the necessary changes in the manuscript has been incorporated and the same has been highlighted in the text in blue.

10. The authors should comment on the limitation of the study as definite conclusion cannot be drawn from such small population of cases.

Response: As suggested we have incorporated the limitation of the study and also have modified our conclusion accordingly and the same has been highlighted in blue in the text.

---

## [Decision Letter · Decision Letter 1]

7 Apr 2020

PONE-D-19-32000R1

Exploring the p53 Connection of Cervical Cancer Pathogenesis Involving North-East Indian Patients

PLOS ONE

Dear Dr. Husain,

Thank you for submitting your manuscript to PLOS ONE. After careful consideration, we feel that it has merit but does not fully meet PLOS ONE’s publication criteria as it currently stands. Therefore, we invite you to submit a revised version of the manuscript that addresses the points raised during the review process.

We would appreciate receiving your revised manuscript by May 22 2020 11:59PM. To enhance the reproducibility of your results, we recommend that if applicable you deposit your laboratory protocols in protocols.io, where a protocol can be assigned its own identifier (DOI) such that it can be cited independently in the future. For instructions see: http://journals.plos.org/plosone/s/submission-guidelines#loc-laboratory-protocols

We look forward to receiving your revised manuscript.

Kind regards,

Kalimuthusamy Natarajaseenivasan

Academic Editor

PLOS ONE

Reviewers' comments:

Reviewer's Responses to Questions

**Comments to the Author**

1. If the authors have adequately addressed your comments raised in a previous round of review and you feel that this manuscript is now acceptable for publication, you may indicate that here to bypass the “Comments to the Author” section, enter your conflict of interest statement in the “Confidential to Editor” section, and submit your "Accept" recommendation.

Reviewer #1: All comments have been addressed

Reviewer #2: All comments have been addressed

2. Is the manuscript technically sound, and do the data support the conclusions?

Reviewer #1: Yes

Reviewer #2: Partly

3. Has the statistical analysis been performed appropriately and rigorously? 

Reviewer #1: Yes

Reviewer #2: No

4. Have the authors made all data underlying the findings in their manuscript fully available?

Reviewer #1: Yes

Reviewer #2: No

5. Is the manuscript presented in an intelligible fashion and written in standard English?

Reviewer #1: Yes

Reviewer #2: Yes

6. Review Comments to the Author

Reviewer #1: In the manuscript entitled “Exploring the p53 Connection of Cervical Cancer Pathogenesis Involving North-East Indian Patients” the authors have discussed the mutation and expression aspects of p53 in cervical cancer pathogenesis

The authors have satisfactory modified the manuscript. However, few minor concerns need to be addressed before final acceptance.

Materials and methods:

In: HPV screening and genotyping by PCR

“using the specific primers for HPV16 1nd HPV” ???..Please clarify… “1nd”

I am failing to understand that, when authors have performed HPV16 and 18 detection using type specific primers, then why the results pertaining to HPV16 and 18 frequencies have not been incorporated the results section. If at all cases were HPV16 positive than it should be highlighted in the results

Moreover, the screening results of HIV, Hepatitis’s virus etc in the blood samples should be also highlighted. Further, the authors should clarify “etc” pathogens.

A brief mention of methods of detection of “HIV, Hepatitis’s virus etc” should be made.

Again the spelling of “Josefesson” is wrong in the text. It should read “Josefsson”.

Authors should rigorously check the spelling of all the citations made in the text and reference list.

Reviewer #2: Review Comments

Response to query 2

I feel that the results which are not statistically significant should also be mentioned with p-values in the manuscript or else the information seems incomplete.

Response to query 3

As mentioned by the author in the manuscript “Further the blood samples were used to screen the presence of any other pathogenic infection such as HIV, Hepatitis’s virus etc apart from HPV in the collected samples.” As per the above statement I would like to know whether HPV was tested from the blood samples. If so, please mention how one could assess the presence of HPV infection from blood samples.

Response to query 6

Please include all the results with their statistical value in a single table format for better visualization and understanding.

Response to query 8

As mentioned by the author the antibody used for immunohistochemistry was ab26 (abcam) however in the datasheet of the company it is mentioned that this antibody was not tested for IHC by the company. So, if you could kindly clarify and justify why this antibody was chosen for immunohistochemistry?

Author has mentioned in the response 8 that “The antibody used i.e. ab26 (Abcam, UK) is anti-p53 monoclonal antibody capable of recognizing both mutant forms and wild-type human p53.” So I would like to know how you can differentiate which expression is due to mutant form and which is due to wild type human p53.

7. PLOS authors have the option to publish the peer review history of their article (what does this mean?). If published, this will include your full peer review and any attached files.

Reviewer #1: No

Reviewer #2: No

---

## [Author Response · Author response to Decision Letter 1]

10 May 2020

PONE-D-19-32000R1

Exploring the p53 Connection of Cervical Cancer Pathogenesis Involving North-East Indian Patients

PLOS ONE

Dear Dr. Husain,

Thank you for submitting your manuscript to PLOS ONE. After careful consideration, we feel that it has merit but does not fully meet PLOS ONE’s publication criteria as it currently stands. Therefore, we invite you to submit a revised version of the manuscript that addresses the points raised during the review process.

We would appreciate receiving your revised manuscript by May 22 2020 11:59PM. To enhance the reproducibility of your results, we recommend that if applicable you deposit your laboratory protocols in protocols.io, where a protocol can be assigned its own identifier (DOI) such that it can be cited independently in the future. For instructions see: http://journals.plos.org/plosone/s/submission-guidelines#loc-laboratory-protocols

A rebuttal letter that responds to each point raised by the academic editor and reviewer(s). This letter should be uploaded as separate file and labeled 'Response to Reviewers'.

A marked-up copy of your manuscript that highlights changes made to the original version. This file should be uploaded as separate file and labeled 'Revised Manuscript with Track Changes'.

An unmarked version of your revised paper without tracked changes. This file should be uploaded as separate file and labeled 'Manuscript'.

We look forward to receiving your revised manuscript.

Kind regards,

Kalimuthusamy Natarajaseenivasan

Academic Editor

PLOS ONE

Reviewers' comments:

Reviewer's Responses to Questions

Comments to the Author

1. If the authors have adequately addressed your comments raised in a previous round of review and you feel that this manuscript is now acceptable for publication, you may indicate that here to bypass the “Comments to the Author” section, enter your conflict of interest statement in the “Confidential to Editor” section, and submit your "Accept" recommendation.

Reviewer #1: All comments have been addressed

Reviewer #2: All comments have been addressed

2. Is the manuscripts technically sound, and do the data support the conclusions?

Reviewer #1: Yes

Reviewer #2: Partly

3. Has the statistical analysis been performed appropriately and rigorously?

Reviewer #1: Yes

Reviewer #2: No

4. Have the authors made all data underlying the findings in their manuscript fully available?

Reviewer #1: Yes

Reviewer #2: No

5. Is the manuscript presented in an intelligible fashion and written in standard English?

Reviewer #1: Yes

Reviewer #2: Yes

6. Review Comments to the Author

Reviewer #1 Comment:

 In the manuscript entitled “Exploring the p53 Connection of Cervical Cancer Pathogenesis Involving North-East Indian Patients” the authors have discussed the mutation and expression aspects of p53 in cervical cancer pathogenesis

The authors have satisfactory modified the manuscript. However, few minor concerns need to be addressed before final acceptance.

Materials and methods:

In: HPV screening and genotyping by PCR

“using the specific primers for HPV16 1nd HPV” ???..Please clarify… “1nd”

Author Response: It was typographical error. Instead of “1nd” it should have been “and” and the necessary correction has been made in the text and the same has been highlighted in blue. We apologize for the mistake

I am failing to understand that, when authors have performed HPV16 and 18 detection using type specific primers, then why the results pertaining to HPV16 and 18 frequencies have not been incorporated the results section. If at all cases were HPV16 positive than it should be highlighted in the results

Author Response: The table has been modified as per the Reviewer’s kind suggestion. As a part of an initial screening process we performed the genotype analysis for both HPV16 and HPV18 and the data showed the presence of HPV16 genotype only in the studied cohort. The line stating the same was previously missing but is now included in the text. The modified table including the details of HPV genotype data has also been included, and the same may kindly be considered at your end.

Moreover, the screening results of HIV, Hepatitis’s virus etc in the blood samples should be also highlighted. Further, the authors should clarify “etc” pathogens.

A brief mention of methods of detection of “HIV, Hepatitis’s virus etc” should be made.

Author Response: As suggested we have provided the brief description for detection method for pathogenic screening, as well the screening results in the materials and methods and result section of the manuscript.

Again the spelling of “Josefesson” is wrong in the text. It should read “Josefsson”. 

Authors should rigorously check the spelling of all the citations made in the text and reference list.

Author Response: We apologize for the mistake. As suggested the necessary correction has been made in the manuscript and the same has been highlighted in blue.

Reviewer #2: Review Comments

Response to query 2

I feel that the results which are not statistically significant should also be mentioned with p-values in the manuscript or else the information seems incomplete.

Author response: As suggested, we have incorporated the p-values for the statistically insignificant result as well and the same have been highlighted in blue in text. 

Response to query 3

As mentioned by the author in the manuscript “Further the blood samples were used to screen the presence of any other pathogenic infection such as HIV, Hepatitis’s virus etc apart from HPV in the collected samples.” As per the above statement I would like to know whether HPV was tested from the blood samples. If so, please mention how one could assess the presence of HPV infection from blood samples.

Author response: No, the presence of HPV infection was not assessed from blood samples, it was determined from the cervical tissues. The blood samples were used to rule out the presence of any other pathogenic infections (as mentioned in the results section) as the exclusion criteria of our study.

Response to query 6

Please include all the results with their statistical value in a single table format for better visualization and understanding.

Author response:

The difference in the relevant p53 data at all levels and their statistical differences have been either illustrated in the figures 1, 2A, 2B and 4C as well as in Table 3, or have been encrypted in the text under different results sections. So we feel that compiling the data set under one common table will be a duplication of data representation, and also make the data set quite complex (as different parameters results have been discussed under different sub-sections under the results. But based on the suggestion, we have now tabulated the p53 exon 4 genotyping data as supplementary table for ready reference and consideration at your end.

Supplementary Table: Details of p53 exon4 polymorphism based difference in genotype analysis 

Stages HPV +ve (N=72) HPV -ve (N=13)

 N Pro/Pro Arg/Pro or Arg/Arg P value ODDs ratio N Pro/Pro Arg/Pro or Arg/Arg P value ODDs ratio

IIA and IIB 48 9 [18.75] 39 [81.25] ref 0.877 (0.258-2.979) 09 5 [55.55] 4 [44.45] ref 1.250 (0.118- 13.240

IIIA, IIIB

and IV 24 5 [20.83] 19 [79.17] 0.834 04 2 [50.00] 2 [50.00] 0.859 

Response to query 8

As mentioned by the author the antibody used for immunohistochemistry was ab26 (abcam) however in the datasheet of the company it is mentioned that this antibody was not tested for IHC by the company. So, if you could kindly clarify and justify why this antibody was chosen for immunohistochemistry?

Author has mentioned in the response 8 that “The antibody used i.e. ab26 (Abcam, UK) is anti-p53 monoclonal antibody capable of recognizing both mutant forms and wild-type human p53.” So I would like to know how you can differentiate which expression is due to mutant form and which is due to wild type human p53.

Author response: In total two anti-p53 antibodies were used for p53 expression studies based on IHC. The first antibody was specifically for IHC, but didn’t yield the expected acceptable results. The second anti-p53 antibody (ab26, Abcam) was chosen for its capabilities to recognize both mutant and wild type conformational forms of p53. Although Abcam (Abpromise guarantee) states that it covers the use of ab26 in IP, ICC/IF, WB; but didn’t specified that it cannot be used for other applications like IHC. We therefore experimentally performed and validated the results with the ab26 p53 antibody by IHC under the supervision of a senior pathologist. Henceforth, the utility of the ab26 p53 antibody for IHC was also demonstrated experimentally. Unfortunately, one cannot claim directly that the p53 protein expression was based on which conformational form of p53 (as ab26 detects both the conformations), but it could be done in association with the genotype data. But, as the post-transcriptional modifications and regulations of p53 are now been realized as an important aspect and complex (outside the purview of the present work), therefore we have not commented on the association of p53 protein expression with the p53 genotype data (only the association of p53 mRNA data with the p53 genotype data was analyzed and documented).

---

## [Decision Letter · Decision Letter 2]

19 Aug 2020

Exploring the p53 Connection of Cervical Cancer Pathogenesis Involving North-East Indian Patients

PONE-D-19-32000R2

Dear Dr. Husain,

We’re pleased to inform you that your manuscript has been judged scientifically suitable for publication and will be formally accepted for publication once it meets all outstanding technical requirements.

Kind regards,

Kalimuthusamy Natarajaseenivasan

Academic Editor

PLOS ONE

Additional Editor Comments (optional):

Reviewers' comments:

Reviewer's Responses to Questions

**Comments to the Author**

1. If the authors have adequately addressed your comments raised in a previous round of review and you feel that this manuscript is now acceptable for publication, you may indicate that here to bypass the “Comments to the Author” section, enter your conflict of interest statement in the “Confidential to Editor” section, and submit your "Accept" recommendation.

Reviewer #1: All comments have been addressed

Reviewer #2: All comments have been addressed

Reviewer #3: All comments have been addressed

Reviewer #4: All comments have been addressed

2. Is the manuscript technically sound, and do the data support the conclusions?

Reviewer #1: Yes

Reviewer #2: Partly

Reviewer #3: Yes

Reviewer #4: Yes

3. Has the statistical analysis been performed appropriately and rigorously? 

Reviewer #1: Yes

Reviewer #2: Yes

Reviewer #3: Yes

Reviewer #4: Yes

4. Have the authors made all data underlying the findings in their manuscript fully available?

Reviewer #1: Yes

Reviewer #2: Yes

Reviewer #3: Yes

Reviewer #4: Yes

5. Is the manuscript presented in an intelligible fashion and written in standard English?

Reviewer #1: Yes

Reviewer #2: Yes

Reviewer #3: Yes

Reviewer #4: Yes

6. Review Comments to the Author

Reviewer #1: (No Response)

Reviewer #2: Response to query 8 is not satisfactory.

As mentioned in the datasheet of the Antibody ab26 (abcam) under specificity that this antibody “recognizes an epitope that is structurally hidden in the wild type conformation of p53 but becomes exposed by denaturation or in mutant conformations of p53 where point mutations in the TP53 gene alter the structure of the protein”. Thus, I feel that the following author’s statement in result section “Similar to mRNA expression the expression of p53 protein was down-regulated in lower severity grade (≤ stage III) compare to higher severity grade (≥stage III) indicating its role in the initial stage of development of cervical cancer” is a bit controversial as this protein expression result by IHC with this antibody could also be interpreted as that the p53 mutation was more in higher severity grade and less in lower severity grade. So I think mRNA expression in this case cannot be supported by IHC with this antibody as stated in this paper rather it makes the result and discussion controversial.

I would rather recommend omitting the IHC part or rewriting the result and discussion accordingly or redo the IHC part with more appropriate antibody.

Reviewer #3: (No Response)

Reviewer #4: The authors have addressed all the issues raised by other reviewer and the manuscript is suitable for publication.

7. PLOS authors have the option to publish the peer review history of their article (what does this mean?). If published, this will include your full peer review and any attached files.

Reviewer #1: No

Reviewer #2: No

Reviewer #3: Yes: Dr Divya Uppala

Reviewer #4: Yes: Amit Kumar Pandey

---

## [Editor Report · Acceptance letter]

9 Sep 2020

PONE-D-19-32000R2 

Exploring the p53 Connection of Cervical Cancer Pathogenesis Involving North-East Indian Patients 

Dear Dr. Husain:

I'm pleased to inform you that your manuscript has been deemed suitable for publication in PLOS ONE. Congratulations! Your manuscript is now with our production department. 

Kind regards, 

on behalf of

Dr. Kalimuthusamy Natarajaseenivasan 

Academic Editor

PLOS ONE